# On Power Line Positioning Systems

**DOI:** 10.3390/s22207827

**Published:** 2022-10-14

**Authors:** Lisandro Lovisolo, Fernando Cruz-Roldán, Manuel Blanco-Velasco

**Affiliations:** 1Department of Electronics and Communications Engineering, Rio de Janeiro State University (UERJ), Rio de Janeiro 999074, Brazil; 2Department of Signal Theory and Communications, University of Alcalá (UAH), 28871 Alcalá de Henares, Spain

**Keywords:** positioning, power line positioning system, power line communications

## Abstract

Power line infrastructure is available almost everywhere. Positioning systems aim to estimate where a device or target is. Consequently, there may be an opportunity to use power lines for positioning purposes. This survey article reports the different efforts, working principles, and possibilities for implementing positioning systems relying on power line infrastructure for power line positioning systems (PLPS). Since Power Line Communication (PLC) systems of different characteristics have been deployed to provide communication services using the existing mains, we also address how PLC systems may be employed to build positioning systems. Although some efforts exist, PLPS are still prospective and thus open to research and development, and we try to indicate the possible directions and potential applications for PLPS.

## 1. Introduction

Positioning systems aim to estimate where a target is. Meanwhile, power line infrastructure is available almost everywhere. Consequently, it may be advisable to use power lines for some applications requiring positioning estimates in some scenarios and environments. This work reports how power lines and Power Lines Communication (PLC) systems may be employed opportunistically to design systems capable of returning position estimates.

Positioning applications include the localization and tracking of persons and lost objects or assets in industrial buildings as well as context- and position-dependent data forwarding as in museums and guided tours, to name a few [1,2,3]. The required positioning accuracy is highly dependent on the application and technology, since the position estimate can be a region the target is, its locality, its coordinates, or the relative distance to a reference (ranging from meters to millimeters). Positioning systems have been designed aiming at global coverage, the so-called Global Positioning Systems (GPS), and to operate inside buildings, the so-called Indoor Positioning System (IPS). Although in outdoor scenarios, GPS technologies are the de facto standard, technology development for cost-effective IPS is still ongoing [1,2,3,4]. Different technologies and strategies have been employed. The target device (or tag) to be located normally collects data from anchors to resolve the positioning; however, systems that collect data from the target or its effects also exist. Some positioning systems opportunistically employ references from legacy systems—as mobile communication (GSM, 3G, 4G), local and personal area networks (WiFi, Bluetooth, UWB), and Radio-Frequency Identification technologies, or also from emerging communication systems such as Visible Light Communication (VLC), to name a few—to resolve the position.

Positioning systems employ different physical quantities and techniques within different coverage areas [1,2,3,5,6,7,8,9,10,11,12,13,14,15,16,17,18,19,20,21,22,23], as shown in Figure 1. In the figure, we separate the positioning systems into global and local. The global ones are designed to operate almost worldwide. Meanwhile, the local ones return a relative position to a local reference, since the operating principles and their interplay with constructions and other objects generally lead to different constraints that restrict their usage.

Figure 1 also presents (though not exhaustively) the different physical principles and quantities employed and the corresponding sensors for collecting the positioning features. Some features are artificial, while others are nature-born. In this work, we discuss the possibilities of obtaining positioning features from the power line infrastructure.

We borrow the name power line positioning system (PLPS) from [24] but expand it to include any system relying on the electrical wiring, the environmental effects of the energy and signals it may conduct, or the effects of persons on signals flowing in the lines to obtain position estimates. PLPS might be easier to deploy and are less visible than radio-frequency (RF) counterparts. Wiring placement is more stable over time than RF access points disposition; consequently, the databases used in some PLPS may be up to date longer than those built for RF signals-based positioning. Complementary, Power Line Communication (PLC) systems [25] are being deployed to support different applications in several scenarios, presenting an opportunity for designing PLC-based IPS (PLC-IPS).

There are many tutorials, surveys, reviews, and books on positioning systems [1,5,6,13,14,17,26,27] and on the use of electrical wiring for communications [25,28,29,30]. Nevertheless, they do not delve into using power lines for positioning. This article’s main contribution is presenting how the power line infrastructure can be used for positioning. We survey the different efforts, working principles, implementation possibilities, and application scenarios that could benefit from PLPS and PLC-IPS. We analyze the positioning references that can be derived from or inserted in the power lines. We provide a systemic categorization of PLPS based on the interaction between the target and the remaining components of the positioning system. In this sense, the use of positioning infrastructure equipment or a target that helps for resolving the position is discussed. For each category, we summarize existing efforts, the possible features for computing the positioning, the techniques to acquire them, and the methods to resolve the target position.

Section 2 discusses general aspects of PLPS and introduces some classification dimensions. The systematic analysis of the use of the electrical wiring, mains, PLC, and the system characteristics, operation principles, and configurations for positioning lead to three eclectic PLPS categories. The Infra-less Target-based PLPS are presented in Section 3. The target employs the mains for self-positioning directly without interaction or injection of positioning reference signals (beacon) in the mains, i.e., without any specific PLPS infrastructure. The next broad category is the Infra-based Target-obtuse PLPS presented in Section 4. The power lines are used to deploy unobtrusive positioning systems deploying the adequate PLPS infrastructure without any support from the target. Then, Section 5 discusses the general class of Infra-based Target-based PLPS. It rests on an infrastructure to inject positioning beacons into the mains that are irradiated or relayed for the target to extract features for positioning. Despite the many positioning technologies, some applications and operation scenarios impose defying aspects that may require new positioning technologies. We discuss the problems in each system category and how PLPS can be used in real scenarios in Section 6. Section 7 concludes this survey and prospective article on the use of power lines and power line communications for positioning.

## 2. Power Line Positioning Systems: Characteristics, Properties and Categories

To the extent of our knowledge, the use of the electrical wiring to irradiate specific positioning purpose signals can be traced to two predecessors [24,31]. In [31], the power lines irradiate a synchronization beacon for an acoustic positioning system. Differently, in [24], the power lines are directly used to irradiate tones that are received using a loop antenna, and the sensed tones’ amplitudes are taken as features for fingerprinting positioning.

Autonomous robots may plug into standard, unmodified electrical outlets for recharging, extending the robot autonomy and displacement range, and dispensing particular bases/outlets. The robot must master several movements for plugging in usual outlets, but first off, it must search and find the power outlet. Vision can be used to identify the outlets. However, alternative approaches may rely on sensing the Electromagnetic Field (EMF) generated by the AC fundamental frequency to detect and align to the outlet [32]. Similarly, wireless power transfer maximization depends on correct alignment [33,34].

Using power lines for positioning in an obtrusive manner, more recently, ref. [35] utilizes the effects of electromagnetic coupling on a wave propagating through the electric wiring for occupancy detection (pinpoint the room region a person is from the effects of its body).

The monograph [6] classifies power line-based IPS within “Infrastructure Positioning Systems” utilizing the existing building infrastructure directly or embedding the IPS into the building, hiding the system from its users. Therefore, since a PLPS benefits from the installed wiring, it is infrastructure-dependent. IPS technologies may be building dependent or independent [27]. Building independence refers to disregarding any particular hardware or knowledge of the building for the IPS to work. If building-dependent, the technology employs prior knowledge of the building or some technology existing inside it for resolving the position.

To analyze the different characteristics and possibilities of PLPS, we need to analyze relevant system properties: (i) the introduction of positioning beacons and, subsidiary, the infrastructure that introduces them to make the PLSP operational; (ii) the measurements leading to the positioning features and who accomplishes them; (iii) if the computation of the position estimate is performed at the target or the infra and if there is a collaboration between the target and the PLPS infra to resolve the position. Although some of these characteristics are not exclusive, they can be used to define the three PLPS categories in Table 1.

### 2.1. Positioning Reference Signals

If the PLPS employs reference signals irradiated either intentionally or not by the mains for positioning purposes, we refer to it as Infra-based. The injection of positioning beacons in the mains requires specific drivers and coupling devices, which is an infra for the PLPS. Alternatively, the PLPS may rest on seeing the parts of the mains or sensing the effects of the energized power lines without using a positioning-purpose reference signal. Consequently, an Infra-less (system) refers to PLPS relying solely on the electrical wiring and its parts.

Due to the increasing use of PLC, it appears as a seamless candidate for providing positioning references. Thus, PLC-IPS refers to PLPS utilizing the PLC signals for an IPS. Nevertheless, the large-wavelength and low-intensity EMF is challenging to detect; thus, alternatively, PLC to wireless (PLC/RF) or PLC to light (PLC/VLC) hybrid links may help [36,37,38,39,40,41]. The anchor emitting the reference signal is connected to the outlet or the light socket to up-convert and irradiate the positioning signal conveyed by the PLC; the position computation may rest on many different methods used by wireless- and light-based positioning. Similarly, some PLC subcarriers can excite the magnets for magnetic-ranging positioning.

### 2.2. Collecting the Features

In some PLPS, the target can collect the positioning features from the reference signals, the environment, or other contextual data to compute its position. In this case, we say that the PLPS is Target-based. Otherwise, if the PLPS infrastructure acquires the positioning features, we say it is Target-obtuse.

### 2.3. Computing the Position Estimate

The position estimate can be determined either at the target or infra. Notwithstanding, it is common to envision cooperative Infra-based Target-assisted (Infra-Target cooperation) approaches when the target collects the features (Target-based) and sends them to a positioning server where the target position is estimated. In the case that the position estimate is computed by the target without specific PLPS infrastructure, we say it is an autonomous or self-positioning system. On the other hand, if the PLPS infrastructure computes the target position without target support, we say the system is unobtrusive.

### 2.4. System Categories

Using the discussed properties, three different PLPS categories follow, which are shown in the first column of Table 1. In Infra-less Target-based (ILTB) PLPS architectures, discussed in more detail in Section 3, the target employs the mains without collaboration or support from other infrastructure to resolve the position. Thus, we refer to this paradigm as autonomous or self-positioning.

The Infra-based Target-obtuse (IBTO) PLPS architecture is discussed in more detail in Section 4. It resolves the target position from the target effects on position-purpose reference signals flowing through the wires. If the target is unaware of the PLPS, we say the PLPS is unobtrusive.

The Infra-based Target-based (IBTB) PLPS architecture is discussed in more detail in Section 5. The electrical cables are used to irradiate positioning beacons or transport them for irradiation by an auxiliary infrastructure. The target acquires the positioning features from the reference signals. Any IBTB PLPS is transparent, since the target is aware of the PLPS infrastructure.

### 2.5. PLPS Examples

Table 2 lists the different systems and experiments regarding PLPS design. The works reported in Table 2 indicate possible positioning features from power lines collected using different techniques for different perspectives. The first column refers to the purpose of the positioning system (its application or service) presented in the reference in the last column. The second column maps the work in the three PLPS categories previously presented. Works that are not explicitly PLPS related are indicated using “none”; they are included in the table, since they present experiments and results that may push forward the design and deployment of PLPS. The third column describes the originating physical principle for the feature. The fourth column presents the feature source, which is followed by how it is collected. The sixth column reports the measurement distance (reach) for feature collection, and the seventh presents how the feature is employed for obtaining the position estimate. The eighth column links the feature bandwidth to the working bands of the PLC standards (see Appendix A).

In the following Section 3, Section 4 and Section 5, we discuss the role of the power line infrastructure in different power line positioning technologies. We present examples of systems in each category. We discuss the positioning references or features that can be employed. Part of the features is naturally provided by the power lines; this can lead to infra-less positioning systems. Meanwhile, infra-based PLPS intentionally insert signals in the power lines to support positioning. For the three system categories, we also discuss how to collect the positioning features and possible approaches to resolving the target position.

## 3. Infra-Less Target-Based (Self) PLPS

Figure 2 illustrates the Infra-less Target-based PLPS architecture. The autonomous device senses the environment searching for position clues related to the electrical wiring for self-positioning. The features can be any information related to the power system. They derive from the mains infrastructure, the irradiated EMF, or images to, for example, recognize the outlets or the light fixtures (although if the light conveys positioning-oriented data, we refer to the PLPS as Infra-based).

### 3.1. Examples

Let us discuss the operation principles of the ILTB PLPS in rows two to seven in Table 2 [32,42,43,44,45].

#### 3.1.1. Autonomous Plugging-In

Autonomous plugging-in requires accurate alignment (relative positioning) between the plug prongs and the outlet socket holes with millimeter-scale precision. In [42], the electric field generated by the outlet is used to sense it. Once an outlet is detected, the robot switches to a vision system for plugging in. In [32], the AC carrier effects guide plug insertion utilizing two separate electrodes, subsequently. First off, a disc of copper foil on the plug front senses the electric field to support coarse alignment with the outlet at mid-range. At a closer range, the ground prong (of a smaller cross-sectional area than the copper foil) is used to measure the electric field for plugging in. A scanning procedure that maximizes the field strength is employed separately in both the vertical and horizontal directions for alignment. These systems’ characteristics are listed in rows two to five of Table 2.

#### 3.1.2. Autonomous Pearching

Self-positioning for Unmanned Aerial Vehicle (UAV) perching to power lines may provide ubiquitous recharging power sources and extend the autonomy of an UAV [43,44]. Such initiatives are listed in the sixth row of Table 2. The authors in [43,44] claim that the near magnetic field of transmission lines in the medium voltage section can provide sufficiently accurate and low latency measurements for the UAV to estimate its relative position and pearch. Similar concepts may apply to unmanned vehicles in multistory parking garages using magnetic field measurements [55].

#### 3.1.3. AC EMR Fingerprinting Positioning

Inside buildings, steel and reinforced concrete structures and the different electric sources produce both static and extremely low-frequency magnetic fields [56]. If the magnetic field pattern inside a building has sufficient local variability and stability over time, one can use it for fingerprinting positioning [56,57,58]. This approach does not require deploying particular infrastructure, and three-axis magnetometers, such as those embarked in mobile phones for geomagnetic sensors, can cope with the measures necessary to collect the position-dependent features.

### 3.2. Positioning Features

The device performing the self-positioning uses clues naturally provided by the power lines for positioning. Figure 3 illustrates the possibilities. Image and video processing techniques can be employed to detect and recognize outlets and lighting fixtures. Magnetic and electric fields are also naturally produced by the AC mains; although, the very low frequency of the mains hardens the detection.

### 3.3. Sensors to Embark

Electronic systems suffer the interference of the AC fundamental; the electric network frequency appears in audio and electrocardiogram recordings [59,60], to place a few examples. The ILTB self-positioning PLPS can use this phenomenon. The target can employ magnetic- or electric-field sensors to perceive the AC EMR [45,61,62]. A microphone, a boosting tank circuit, conductors (measuring in centimeters), and small mics (as the ones embarked on mobile phones) can be used to sense the fundamental EMR and capture the mains’ hums. Many specialized sensors have been developed to sense magnetic fields [63]. As discussed, the target may embark image sensors.

### 3.4. Resolving the Position

As above exemplified, an ILTB PLPS is restricted to using positioning features resulting from the AC EMR amplitude pattern in space, besides imaging.

The intensity of the magnetic field produced by a three-phase single circuit power line is derived in [64]. For a power line consisting of three conductors lying on the plane with an inter-distance of *s* meters (the cables are placed at −s, 0, and *s* on the plane axis perpendicular to the power lines cross-sections), conducting currents *I* (modulus of the phases with angles $-\frac{2\pi }{3}$, 0, and $\frac{2\pi }{3}$), the intensity of the magnetic field at the distance *d* in the direction $\psi_{d}$ (considering the line crossing the three conductors perpendicularly) is
(1)$$\begin{array}{c} B\left(d\right)=\frac{\mu_{0}Is}{2\pi d}{\left(\frac{3d^{2}+s^{2}}{d^{4}-2d^{2}s^{2}\cos\left(2\psi_{d}\right)+s^{4}}\right)}^{\frac{1}{2}}. \end{array}$$

By measuring *B*, one can estimate the distance *d* to the power line for the UAV to pearch for recharging.

Image processing and vision techniques on the visible and non-visible bands of the spectra have presented an astonishing advance in the last decade. These techniques can be used to recognize and extract the distance and relative position from many parts of the power system.

The long wavelength of the EMR resulting from the AC fundamental frequency and its, in general, weak intensity indoors may hamper accurate positioning. However, the authors in [45] report that the spatial variation and temporal stability of the mains EMR promisingly hold up for sub-meter accurate positioning in real time. Fingerprinting positioning methods, also called database correlation or scene analysis, employ position-dependent features, i.e., a database of reference patterns with the associated positions [26]. The target collects a fingerprint that is compared to the reference ones to estimate the target position. This strategy can be employed to use the spatial variation and temporal stability of the AC EMR for resolving the target position.

Figure 4a illustrates how fingerprinting works [26]. During operation, the target fingerprint is collected, and the scene analysis algorithm returns position candidates by comparing the target $f_{\mathrm{target}}$ to the reference $f_{k}$ fingerprints. A feature vector $f_{k}$ is collected at position $p_{k}$, *K*; such tandems compose the database of fingerprints, as illustrated at the left of Figure 4b. The scene analysis returns the estimates position
(2)$$\begin{array}{c} {\hat{p}}_{\mathrm{target}}=p_{k}\mathrm{with}k=\arg_{k=\ldots K}\max\mathrm{similarity}\left(f_{\mathrm{target}},f_{k}\right). \end{array}$$

There are many different algorithms for making this comparison and obtaining position candidates, and one may select the position associated with the best match among the reference fingerprints or combine (average/interpolate) the reference position of ranked matches.

Figure 4b depicts the database construction (preparation or off-line) stage when the reference fingerprints, the feature vectors, are collected at known positions. It is common to divide the operating region into cells of size Δx×Δy, associating a fingerprint to one cell—each cell corresponds to a small locality. For the fingerprint, one makes different measurements to confront the inherent variability and stores a few examples, their average over space and time, or the empirical distribution (or parameters modeling it). The overlap of the two stages in Figure 4a indicates that the database can be updated during operation to improve the system’s accuracy and reliability if the working environment requires keeping the databases up to date and valid to work properly.

The authors in [45] present an akin approach, capturing the AC fundamental EMR and inputting it into a fusion-based Simultaneous Localization and Mapping (SLaM) system. Line seven of Table 2 lists this system’s characteristics. The power network EMR pattern across space is combined with a dead-reckoning module. (Velocity is the rate of change in spatial position, and acceleration is the rate of velocity change. From the velocity, one may compute the position change/displacement, and from the acceleration, one may compute the velocity. Movement and inertial navigation systems compute the displacement using the laws of motion [7]. Nowadays, micro-electromechanical systems are embarked on different objects to measure the movement-related physical quantities used to estimate the trajectory and give the relative displacement to the trajectory starting point. These systems may also employ a compass as a complementary datum. Such methods are generally called dead reckoning: the device determines its current position from the past one and its trajectory.) To compute the route, ref. [45] uses a dead-reckoning module, but it employs the received AC fundamental EMR to detect loop closures in the route, in a fingerprinting fashion, recognizing if a point is being revisited.

The ILTB PLPS operational principles discussed in this section rely on sensing the AC mains effects or “seeing” the mains infrastructure. The target is responsible for acquiring the features and computing the position estimate without any additional PLPS infrastructure; besides, the power system is unaware of generating the positioning references. However, it is possible to foresee that there may be data exchange with a server (sending the measurements or querying databases) for the position computation.

### 3.5. Applications

The ILTB self-positioning PLPS may extend the autonomy of robots and electric vehicles. It seems reasonable to include this ability with the many other capabilities envisioned for these machines. Other appliances, such as autonomous vacuum cleaners, could also benefit from this ability. The same concept applies to different machines used in industries and underground mines, to name a few other application domains.

## 4. Infra-Based Target-Obtuse (Unobtrusive) PLPS

Figure 5 depicts the basic architecture for the Infra-based Target-obtuse PLPS discussed in this section. The PLPS infrastructure injects signals into the wiring and collects the features at different points. The electromagnetic coupling between the human body and the wiring affects wave propagation in the wires. The wiring and the coupling between the lines and the Earth change much slower than a person moves. Consequently, the PLPS rests on detecting coupling effects changes that are most likely due to humans. Notice that the person might be unaware of the IBTO PLPS; i.e., it may unobtrusive. Such systems may be employed for simple detection to indicate if a room or locality is occupied.

### 4.1. An Example

The IBTO PLPS is illustrated by the example in the eighth row in Table 2. The work [35] injects a 70 MHz reference into the neutral wire and measures its peak value at five outlets. These features are used to detect and locate the person using a support vector machine. The classifier is trained using features collected with the person positioned among 1.2 m × 1.2 m squares in a 9.6 m^2^ room, achieving an accuracy of 91% accuracy and the authors report that mis-classification increased the closer the person was to the near-field region and that tiny movements of the occupants led to significant changes in the sensed tone peak value. Since only persons are expected to move within short periods, filtering out ambient factors (furniture, equipment, vibration, climate) that can make the mean of the peak value wander over long-term measurements may improve performance [46].

### 4.2. System Model

The IBTO PLPS rests on injecting reference signals into the electrical wiring and identifying the distortions they suffer due to the effects on human bodies. In addition to the injection of the reference signals, the IBTO PLPS infrastructure comprises sensors to measure, collect the features and communicate them. We note that PLC technology could support that.

Figure 6 presents a simplified model for IBTO systems. *N* reference signals are injected at *N* different points. They arrive at *M* detection points. If $h_{nm}$ is the channel response from the *n* reference-signal insertion point to the *m*th detection point, one has
(3)$$r_{m}\left(t\right)=\sum_{n=0}^{N-1}h_{nm}\left(t\right)*s_{n}\left(t\right),$$
where there are NM channels. In the diagram in Figure 6, the references are forwarded to the Positioning Server to resolve the position of the person. The reference signals can be configured by the Controller which decides their frequencies, modulations, and time-division multiplexing.

#### Using PLC for IBTO PLPS

The above model is valid for PLC signals. The simplified model for a PLC symbol is
(4)$$s\left(t\right)=\sum_{c=0}^{C-1}a_{c}e^{j\frac{c2\pi }{C}},$$
where *C* is the number of subcarriers, *c* is the subcarrier index, and $a_{c}\in C$ is the symbol conveyed by the *c*th subcarrier and depends on the constellation applied to modulate it. The Appendix A contains a few details. Thus, using subcarriers from the PLC-OFDM symbol of appropriate frequencies, the changes in the channels due to the coupling between the lines and persons moving in the target area can be detected.

### 4.3. Positioning Features

From the reference signals, one extracts the positioning features to resolve the positioning. They are expected to depend on the person. If one assumes that one can separate the sources, which can be achieved by time or frequency multiplexing, the set of positioning features for the person at a position *k* is given by
(5)$$\begin{array}{c} f_{k}={\left\{\mathrm{features}(h_{nmk}\left(t\right))\right\}}_{n=0\ldots N-1,\;m=0\ldots M-1}. \end{array}$$

If each reference signal is designed to provide *P* features, then at best, there are NMP positioning features, thus
(6)$$\begin{array}{c} f_{k}={\left\{{\left[f_{nmk}^{p}\right]}_{p=0\ldots P-1}\right\}}_{n=0\ldots N-1,\;m=0\ldots M-1}, \end{array}$$
where ${\left[f_{nmk}^{p}\right]}_{p=0\ldots P-1}$ is a vector of features $f_{nmk}^{p}$ derived from $h_{nmk}\left(t\right)$.

Note that simple features can be derived using a set of carriers in each injection point and measuring their amplitudes at the different sensing points. In principle, more carriers increase the capacity to detect the person and discriminate where she is. The insertion points for the tones and their injection powers affect the cover of the IBTO PLPS and its operational area in a given building. The more carriers and injection and sensing points, the more resilient the IBTO is expected to be due to the possible diversity gain.

### 4.4. Resolving the Position

One may apply anomaly/change detection techniques to the features for alarming the presence or movement of a person.

If finer locality information is desired, it is perhaps necessary to acquire prior information about the operating environment and the effects of persons on it, i.e., utilize fingerprinting. An offline preparation phase must occur to learn the signal features with and without the person coupling effects at different positions. So that, after, during operation, they can be associated with different positions.

### 4.5. Applications

The PLPS discussed in this section rests on the coupling among humans and wires to detect the person’s presence and provide her locality indication.

The ubiquity of wireless networks led to the concept of passive positioning systems [65]. The active systems employ tags/electronic devices in the target to be tracked [66] that assist in resolving the position—they are Target-based. In opposition, sensors are deployed in passive systems to capture the possible effects of the target presence. Such systems have been used to detect people [65,67], to resolve their localities (and ultimately the positions) [65,68,69,70], and to assess the individuals in a given region [68]. Such human-presence detection systems are device-free because the target does not carry a sensor. The device-free system is unobtrusive, since it does not disturb the persons being located that could even be unaware of it. As we have seen, the IBTO PLPS may passively detect a person and estimate her locality.

Locality Classification (LC) answers if a target (object or person) is within a given site [71]; if it is inside/outside a region, and, in some cases, the area in the region where it is. Figure 7 illustrates the concept. The upper-left corner illustrates LC in an office. The locality may range from the coarser indication of the row of booths (top or bottom, as illustrated by the colors), the column (the different patterns), and the finer indication of the booth (the blending of colors and patterns). The down-right corner illustrates the LC of road segments according to the light pole providing the illumination (as indicated by the different colors). The size of the regions depends on the number of references, their powers, and the propagation environment. Although LC may present a small resolution, it is expected to present less error than reporting coordinate estimates.

Considering the appropriate constraints, the IBTB PLPS that aims for presence detection and person LC can operate in office and residential buildings, houses, hospitals, nursing homes, and other facilities. It can be used for detection, alarming, or pinpointing the presence of persons (interns, patients, and elders) in a given room to reduce hazards [67,72,73,74]. Such cognizance of the building usage and occupancy helps to understand and manage energy usage [20] and provides context information for controlling electric appliances in smart homes [75,76,77]. Although it employs a specific infrastructure, it can operate independently of the target knowledge of its existence and people’s agreement and be unobtrusive. Consequently, IBTO PLPS also enables intruder detection, substituting many acoustic (ultra-sound) devices and infrared detectors and cameras deployed to detect people’s presence in rooms.

## 5. Infra-Based Target-Based PLPS

Figure 8 depicts the basic architecture for Infra-based Target-based PLPS. The target senses the positioning reference signals injected into the power line by the IBTB PLPS infrastructure. The target collects the features that are used to estimate its position. The target can estimate the position itself, or another entity can resolve it.

### 5.1. Examples

The IPSs in [24,47], rows nine and ten in Table 2, employ infrastructure modules to inject positioning purpose tones in the building’s electrical wiring at different locations. The tones (sinusoidal waves) traveling through the wiring produce EMR. The amplitude and phase of the EMR resulting from tones are expected to vary in space according to the building blueprint, the wiring itself (placement, density, and other characteristics), and the tone injection point. If the target measures the amplitudes and phases of the tones, its position could be resolved using fingerprinting. The viability of such a scheme depends on acquiring the EMR produced by the tones, and the techniques depend on the frequency of the reference tones. Section 5.5 below brings a few possibilities.

In principle, using more tones and injecting them at more points increases the reliability and diversity of the features and, consequently, the positioning accuracy. Therefore, refs. [47,78] extend the two tones (30 kHz and 447 kHz—i.e., in the 10–500 kHz) used in [24,79] to 44 tones (ranging from 447 kHz to 20 MHZ). For cells with 3–4 m sides, the two-tone PLPS achieves room-level accuracy (LC in terms of a room) and sub-room-level accuracy (position inside the room) in the ranges of 78–100% and 87–95%, respectively; and, using a 2–3 m grid for the 44-tones PLPS, the authors report accuracy gains in the range of 12–21%.

### 5.2. System Model

The first IBTB PLPS approach [24] injects specific tones in the mains to be irradiated by the cables. They are detected and sensed using a loop antenna for the fingerprint. This is the basic model for the IBTB PLPS.

The infrastructure for the injection of the reference signals for IBTB PLPS is similar to the one of the IBTO PLPS. This is depicted in Figure 9. The infrastructure devices connect to the wiring to inject signals that provide the references from which the features are obtained by the target for positioning. In comparison with the IBTO, the difference resides on the collection of the features that in IBTB is accomplished by the target.

### 5.3. Positioning Features

In IBTB PLPS, the target must collect the effects of the EMR produced by the injected signal references to resolve the position. With reference to Figure 9, at the position *k*, the collected signal is
(7)$$r_{k}\left(t\right)=\sum_{n=0}^{N-1}{\tilde{h}}_{nk}\left(t\right)*s_{n}\left(t\right).$$

Note that ${\tilde{h}}_{nk}\left(t\right)$ is the hybrid wiring-wireless channel response from the signal origin *n* to the position *k* and is expected to vary with the position. Each ${\tilde{h}}_{nk}\left(t\right)$ accounts for both the wired and the wireless channels from injection point *n* to the spatial point *k* where the EMR is measured, which is the position that the system must obtain. If one assumes that one can separate the sources, which can be achieved by time or frequency multiplexing, the set of positioning features for the target at a position *k* is given by
(8)$$\begin{array}{c} f_{k}={\left\{\mathrm{features}({\tilde{h}}_{nk}\left(t\right))\right\}}_{n=0\ldots N-1}. \end{array}$$

From the collection of ${\tilde{h}}_{nk}\left(t\right)$, one obtains the position (*k*)-dependent features as the impulse response ${\tilde{h}}_{nk}\left(t\right)$ itself, the modulus of the correspondent frequency response ${\tilde{H}}_{nk}\left(f\right)$, its values at defined reference frequencies or tones *c*, where ${\tilde{H}}_{nk}\left(c\right)$, c∈C (C) is the set of tones used for the IBTB PLPS, or the tones phase differences, for example.

The diversity of features depends on using more injection points for reference signals or extracting more features from each signal. Meanwhile, in the IBTO model, one can also extend the number of sensing points to produce more features. Nevertheless, as for the IBTO PLPS, the coverage, reliability, and accuracy of IBTB PLPS depend on the number of reference carriers, where they are injected, and the wires where they flow that may make tones arrive and be sensed at different places in the building; i.e., their coverages differ. Consequently, the more tones and the greater their powers, the more extensive the combined coverage is expected to be, meaning that *k* may index a larger area.

Reliability and accuracy are expected to increase the more tones there are composing the reference signals and the more reference insertion points *n* that are employed, which is due to the better and more consistent measurements resulting from using more carriers and a larger dispersion of the entry points.

We note that different targets may present different measurements ${\tilde{h}}_{nk,\mathrm{target}}$; this cross-device effect can be accounted for introducing a target-dependent gain $G_{\mathrm{target}}$ and computing ${\tilde{h}}_{nk}^{\prime }={\tilde{h}}_{nk,\mathrm{target}}\left(t\right)/G_{\mathrm{target}}$ to obtain the features (although we exemplified only gain target differences, the approach can be extended to consider differences in frequency response as well).

The architecture of IBTB PLPS using the irradiated EMR from signals traveling in the power wiring resembles positioning efforts using the leak of RF signals from coaxial cables [80,81]. There are two correlated differences: the signal bandwidth (much higher and maybe larger) and the cable employed (coaxial). The coaxial cables are prepared to irradiate the RF signal satisfactorily, while IBTB PLPS intends to use the existing mains infrastructure.

### 5.4. IBTB PLPS Using PLC

PLC technology, Appendix A, is available for smart grids, smart cities, smart buildings, Industry 4.0, and IoT services that demand positioning services. Therefore, we analyze how PLC may support IBTB PLPS.

In this case, the signal inserted in each injection point *n* is given by Equation (Equation 4). The PLC signal has a tone-based structure that can favor feature extraction.

PLC modems connect using differential mode currents flowing through the wiring. However, Common-Mode (CM) current arises due to line imbalance [82,83,84,85] producing undesired EMR. As a result, PLC modems must comply with Electromagnetic Compatibility (EMC) regulations that limit their EMR [25,86,87,88,89]. Several works report experiments on the EMR resulting from PLC signals [51,82,83,84,85,90,91]. Nevertheless, one finds some efforts to measure the PLC EMR and characterize the PLCWC due to EMC regulations and concerns about PLC connections eavesdropping.

The EMR resulting from CM propagation of PLC signals presents intensity with varying spatial patterns [84,88]. Consequently, the EMR resulting from PLC may be used to construct an IBTB PLPS, as depicted in Figure 8. This entails acquiring features from the PLC-Wireless Channel (PLCWC), which is the channel resulting from using a PLC modem as a transmitter, while reception depends on detecting the resulting EMR [52,53]. If the target captures the PLC EMR, some subcarriers of the OFDM symbols can encode information for positioning purposes. Notwithstanding, regardless of the PLC standard and operation environment, fingerprinting methods seem to be the most reasonable approach to resolving the target position due to the difficulty in modeling the PLCWC.

### 5.5. Sensing the PLCWC

Techniques to detect and measure and extract the channel response or channel state information (CSI) from the PLCWC are necessary for designing a PLC IPS. Although the frequency band and the wire length elude the far-field assumption, we revisit some efforts that report successful techniques for the characterization of the PLCWC.

One approach to obtain the PLCWC CSI is by means of a loop antenna [47,51,52,84,86,88,91]. For example, in [47], a broadband loop antenna of 1 m in diameter is utilized for sensing 44 tones ranging from 447 kHz to 20 MHZ injected in the power line for an IBTB PLPS using fingerprinting. A 10 cm loop antenna is employed in [51] to characterize the PLCWC inside a building, and [52] uses a 30 cm loop antenna for measurements up to 40 MHz; they conclude that PLC is prone to eavesdropping.

Another approach can be seen in [53], where the authors employ omnidirectional and monopole antennas to measure the PLCWC in the 1.7 to 100 MHz band. They mention experiments inside different buildings and distances between the wiring and the antenna up to 6.0 m. They conclude that in the BB-PLC bandwidth, the PLCWC presents severe attenuation and frequency selectivity.

From the initiatives to characterize the PLCWC and obtain the CSI, as seen in rows 12 to 14 in Table 2, one sees that it is possible to sense the EMR from power line signals in the MB- and BB-PLC frequency bands. As a result, we advocate that it is possible to use the PLCWC for indoor positioning.

### 5.6. Resolving the Position

For any of the above features derived from $h_{nk}\left(t\right)$, fingerprinting (see Section 3.4) can be the means for effectively resolving the position of IBTB PLPS, since it is unfeasible to predict the EMR resulting from carriers traveling through a building cabling.

### 5.7. Hybrid PLPS Using PLC Technology

Hybrid channels may relay the PLC signals to provide the positioning features. Below, we delve into some possibilities.

#### 5.7.1. PLC–IPS Using Magnetic Beacons

Power lines can deliver the signals for magnetic positioning. This system model is as in Figure 9, but instead of using the EMR produced by the reference signals flowing in the wiring, coils are used to irradiate positioning beacons. This scenario is depicted in Figure 10. This can be deployed by using low-frequency PLC carriers to drive the coils producing the magnetic beacons for positioning. The target also embarks coils for sensing the resulting magnetic fields. The ranges or locality identities can be obtained from the detected field.

The intensity of the voltage induced in the target coil depends on the relative position between the target and the anchor. For example, if both coils have the same specifications and orientations, the target–anchor distance follows [49]
(9)$$\begin{array}{c} d=10^{-\frac{\log\left(V_{\mathrm{target}}\right)-\log\left(\frac{\omega \mu_{0}mNr^{2}}{4}\right)}{3}}, \end{array}$$
where $V_{\mathrm{target}}$ is the voltage induced at the target coil, ω is the angular frequency of the carrier, $\mu_{0}$ is the vacuum magnetic permeability, *m* is the modulus of the magnetic moment ($\pi NI_{\mathrm{anchor}}r^{2}$), *N* is the number of turns, $I_{\mathrm{anchor}}$ is the current in the anchor coil, and *r* is its radius. Alternatively, commercial magnetic sensors can also be used to measure the magnetic field.

From the ranges to different anchors, circular multilateration leads to the position estimate, as shown in Figure 11 [2]. The solution is given by the interception of the circles centered at the anchors.

Using this paradigm, ref. [48] employs reference signals around 189 kHz to excite the coils [49], around 24 kHz, and [50], at 125 kHz. These carriers fall within the NB-PLC frequency bands. One or multiple coils (and resonators to extend the operating range) can be used in the anchors and the target, and one or multiple carriers can be produced, simultaneously or sequentially for magnetic-field-based ranging and positioning [12,48,50,92,93,94,95,96].

#### 5.7.2. PLC–Wireless Integration

The Hybrid PLC–Wireless Channel (HPLCWC) is the tandem of wired and wireless links [97,98,99,100]. The PLC signal goes wireless upon the appropriate frequency shift (and the possible rearrangement of the OFDM frame). This system model is illustrated in Figure 9, but instead of using the EMR produced by the reference signals flowing in the wiring, antennas are used to irradiate positioning beacons. This leads to the IBTB PLPS depicted in Figure 12. We highlight that the HPLCWC uses the frequency band employed by wireless communications systems, whereas the PLCWC lies in the PLC frequency band. In this case, the many strategies existing for RF-signal detection and positioning in wireless networks (fingerprinting, multilateration, and multiangulation) can be used [1,2].

#### 5.7.3. PLC–VLC Integration

The VLC technology [101,102,103] turns illumination fixtures into high-capacity downlink connections. The high interference immunity to other electromagnetic sources and opaque walls favor VLC and its channel reuse indoors, respectively.

In [104], lighting fixtures emit identifications for locality classification. If the light conveys the appropriate information, the ambient light can support LC [11,15,16]. Positioning algorithms using light-based ranging, the incident angle of the light, and its intensity can also be employed [11,15,16]. Even though the identity can be assigned directly for each light bulb, it can also be set in a coordinate fashion by the PLC system. Utilizing the PLC–VLC hybrid channel to emit the positioning beacons traveling in the power lines is also possible [105].

Such a hybrid PLC–VLC PLPS presents a basic model as in Figure 9, but instead of using the EMR produced by the reference signals flowing in the wiring, light sources are used to irradiate positioning beacons. The PLC–VLC PLPS is illustrated in Figure 13. Light fixtures emit the beacons, and the target employs photo-diodes or image sensors to sense the VLC channel and collect the positioning features.

### 5.8. Applications

The scenario and strategies discussed in this section encompass IBTB positioning systems that measure features from signal references injected in the electrical wiring and irradiated directly by the power line or through hybrid approaches.

Some experiments listed in Table 2 indicate the possibility of measuring the PLCWC and extracting the CSI for MB- and BB-PLC systems, which, for example, can be used for fingerprinting-based positioning.

Using ranging from magnetic beacons for IBTB PLPS has the advantage that magnetic coupling at low frequencies is reasonably robust to people and changes in room configuration.

The PLC-wireless and PLC–VLC hybrid approaches may benefit from frame structures that simultaneously fit PLC and WiFi or PLC and VLC. To this end, frame structures and transceivers that can jointly be used for positioning and communications in BB-PLC, WiFi (wireless), and VLC systems are necessary to back the utilization of the same processing chain for the different physical channels, resulting in a simpler target design and lowering the cost. Meanwhile, the construction of PLC-IPS using Magnetic Fields in Section 5.7.1 could perfectly use NB-PLC carriers.

The many applications of IPS encountered today may rest on the different IBTB PLPS architectures above discussed.

## 6. Some Comments on the Deployment of PLPS and Application Scenarios

Section 2 discussed the characteristics and properties of the three PLPS architectures. We note that although all PLPS architectures aim at a position estimate, their architectures and possible uses are rather different. Thus, in the previous sections, we have pointed out the different positioning features that the different PLPS categories may rely on, how to collect them, and general use-case scenarios and applications for the three PLPS categories.

While the technology of ILTB PLPS relies solely on the power line infrastructure and the energy it delivers, the IBTO and IBTB use the power lines to irradiate or, in some cases, simply conduct the positioning signals for the irradiation by a magnetic, RF, or light transducer. Since the electrical wiring is not in principle intended for data transfer, the bandwidth for the positioning signals presents severe restrictions. Both the IBTO and IBTB PLPS must comply with Electromagnetic Compatibility (EMC) regulations [25,86,87,88,89] that limit the electromagnetic emissions, as any other equipment. PLC modems are designed to effectively use the wiring for communications and to comply with EMC regulations and thus appear as an adequate means for providing the PLPS infrastructure for IBTO and IBTB PLPS. ILTB PLPS do not introduce any positioning reference into the power line; they are autonomous systems that opportunistically employ any aspect of the existing power system for positioning purposes. Section 3 discussed ILTB systems architectures, possible features, and strategies that may leverage autonomous robots and other self-positioning applications. In an ILTB PLPS, the power lines passively provide the positioning features. Being infrastructure-free, such a system can be deployed in many scenarios, although the positioning accuracy may largely vary depending on the features employed for positioning, and, in many cases, switching among different sensors and features may be necessary for accurate positioning.

The IBTO PLPS turns about the ILTB, since the target is passive. An IBTO PLPS rests on how the target affects the signals traveling through the wire lines. Section 4 discussed possible IBTO systems architectures, features, and strategies that may, for example, leverage the obtrusive positioning of people. However, small regions and the small bandwidth of the power lines may hamper to obtain features of sufficient diversity for accurate positioning, as we describe below in Section 6.1; again, PLC signals may be a reasonable approach to increase feature diversity for effective IBTO PLPS implementations.

Since an IBTB PLPS requires equipment on both sides, at the infrastructure to produce the positioning references and at the target for their acquisition, one may say either that an IBTB PLPS lies between ILTB and IBTO or encompasses both. Section 5 discussed IBTB systems architectures, possible features, and strategies; the main conclusion is that the direct use of PLC signals traveling through the power lines may be very difficult if not unfeasible due to the long wavelength and faint intensity of the EMR produced by the PLC. Consequently, a probable scenario for building IBTB PLPS is using the BB-PLC signals to deliver the references for emission using RF or light in a hybrid fashion. Since magnetic coupling positioning can be precise and accurate and is immune to the effects of the human body, another possibility is using NB-PLC carriers through the wiring to excite coils for magnetic positioning in large storage buildings and alike.

Table 2 brings the reaches for feature collection for the different PLPS initiatives. From the data in the literature and the working principles of the different PLPS architectures, it is possible to speculate on their accuracy as well. ILTB systems depend on the sensors employed and can switch among them; when images are employed, centimeter-level accuracy can be expected; meanwhile, at very close distances from a power outlet, millimetric accuracy can be possible. The development and improvement of PLPS IBTB systems using signals traveling through the power lines may lead to an accuracy of meters or even tenths of centimeters. For hybrid IBTB PLPS, the accuracy depends on the second hop that irradiates the beacons. For magnetic-based positioning, centimeter-level accuracy is expected. Using OFDM symbols for RF-based IBTB PLPS, the accuracy can be as high as for WiFi-based positioning systems. If the light is employed instead, accuracy improvements can be obtained. Regarding IBTO PLPS, distinguishing the target position is hardened due to the nature of the collected features. Consequently, the accuracy may be limited to meters, making IBTO PLPS more suitable for location classification than providing a position estimate.

### 6.1. Unobtrusive PLPS Using PLC

PLC modems are already in the market. The transmitter injects the PLC signal into the wiring, and the receiver detects and demodulates it. Consequently, PLC systems may seamlessly provide the infrastructure for the IBTO PLPS. Since the BB-PLC signals lie within a bandwidth where the human–body–conductor coupling may be important, changes in the channel responses for the OFDM subcarriers of BB-PLC can be used for presence detection and possibly LC and positioning. Since PLC and smart metering using PLC are increasingly deployed [106,107,108,109,110], the technology may already be available at several premises.

### 6.2. Positioning in Ships

In [111,112], the authors claim that accurate and pervasive positioning in cruiser ships improves the response to incidents. For example, it can help evacuation routing to reduce hazards during incidents [113]. Other relevant tasks are detecting and counting the workforce in ships’ bridges for safety watchkeeping [114]. Recently, ref. [115] highlights that positioning systems on shipboards could keep track of passengers to identify areas of high transmission risk of infection in cruise ships. Designing PLPS for shipboards may be favored, as PLC systems have been utilized in ships for different applications, most remarkably for the communication among the different automation systems embarked [116,117,118]. Since power is vertically distributed in cruise ships with a reduced number of rooms per segment [118], an unobtrusive PLPS may help to determine the busiest areas on ships. Nonetheless, in-ships, the power distribution may employ armored or unarmored cables [119], affecting the channel response and the resulting EMR.

### 6.3. Underground Mines

Safety concerns apply to underground mines, and positioning can help to reduce hazards. However, the underground mine is a hard environment for positioning systems [22,120]. Variants of existing IPS technologies have been tried in this scenario, but such an environment hampers using acoustic and RF-based positioning systems. Similar arguments indicate that the same obstacles exist in risky construction sites, which are a defying situation for positioning systems [19].

In mines, power must be delivered for different tasks. The deployed infrastructure can be used to back PLPS. Both IBTO and IBTB PLPS can be used in this scenario. For the IBTO PLPS, the concepts already discussed apply as well in underground mines. For deploying IBTB PLPS in underground mines, both magnetic beacons and hybrid PLC–VLC are of interest to provide ranging-based and locality classification.

### 6.4. Industry 4.0

Industry 4.0 is expected to result from massive sensor deployment, high device connectivity, automation, and real-time data acquisition to evolve from computerized manufacturing to cyber–physical systems (physical objects with embedded software and computing power) collaboration. The manufactured products are expected to become smart [121] with self-management capabilities. The Industry 4.0 paradigm aims at improving the industrial process to [122]: (i) fulfill individual customer requirements profitably, (ii) provide flexibility to rapidly respond to disruptions and failures, (iii) achieve transparent decision making of the manufacturing processes, and (iv) automate processes subject to resource and energy efficiency constraints. Many objectives require positioning services at a large scale and low cost for operators, tools, materials, products, and other objects and parts of the production chain to [18,123,124,125,126,127,128,129]: (i) reduce unnecessary displacements, (ii) optimize the layout and the supply chain, (iii) reduce exposure to material and material handling, (iv) support real-time manufacturing, (v) track the tools and their use to optimize their placement and storage, and (vi) reduce stock and waste; to list a few. The required positioning accuracy for Industry 4.0 services may range from simple presence detection and locality identification up to sub-cm accuracy.

Different cabling technologies are used in factories following many standardized procedures. They may be employed to support PLPS in such premises. The different electrical system components provide visual and heat-pattern clues that can be used for ILTB autonomous positioning. In the manufacturing lines that employ a small working force, one may employ IBTO PLPS to locate the workers. Meanwhile, the environment may be defiant: as more machines are used, their powers increase, and their operating cycles vary over time, among other issues. In such scenarios, the hybrid PLC-based IBTB PLPS may help to provide positioning services using the cables as the backbone for the positioning system.

### 6.5. Hybrid and Fusion PLPS

Power lines can partake in hybrid and fused positioning systems [130,131,132]. For example, one may employ a compass, magnetometers, ambient light, pressure, and data from different RF anchors (mobile networks, WiFi, Bluetooth, ZigBee, among others) for a hybrid fingerprint. In addition, light and magnetic ranging could be mixed for multilateration. Alternatively, fingerprinting position estimates can be combined with dead reckoning. These examples illustrate that different features can be combined for hybrid computation of the position estimates or that fused position estimates can also be approached; increasing the diversity of features and positioning strategies extends the availability, reliability, and accuracy of the positioning system.

## 7. Conclusions

This prospective survey mapped how power lines can support positioning services and some of the potential applications for power line positioning systems (PLPS). We devised three main system architectures or classes depending on whether the system uses specific infrastructure to produce reference positioning signals and whether the target makes measurements to collect the features employed to estimate its position. Infra-less Target-based PLPS architecture can be employed for self-positioning by different devices; Infra-based Target-obtuse PLPS architecture can be employed for obtrusive positioning of persons; and, Infra-based Target-based PLPS architecture can be employed for general positioning applications.

Since Power Line Communication (PLC) systems employ the mains infrastructure for communications, this work has also indicated approaches for using PLC to implement Infra-based PLPS. We mapped the PLC spectral bands appropriate for the IBTO and IBTB PLPS architectures that may foster deploying PLC-based indoor positioning systems (PLC-IPS). The development of PLPS is still ongoing and requires more experiments, proposals, and developments. Strategies, devices, sensors, and methods to employ PLC signals emitted directly by the power lines or using a hybrid medium for effective positioning need to be further investigated. We hope this contribution helps PLPS research and development by providing relevant directions for these tasks.

## Figures and Tables

**Figure 1 sensors-22-07827-f001:**
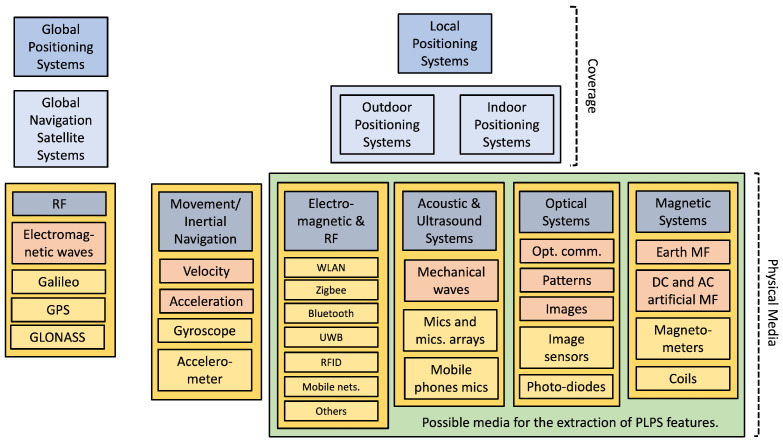
Schematic view of systems and physical quantities used for positioning. Some systems are specially designed for positioning, such as the GPS, while others opportunistically employ legacy systems for positioning. The latter may rely on artificial or nature-born information. The green rectangle indicates the possibilities for PLPS construction—the characteristics, technologies, and applications for PLPS are the subject of this work.

**Figure 2 sensors-22-07827-f002:**
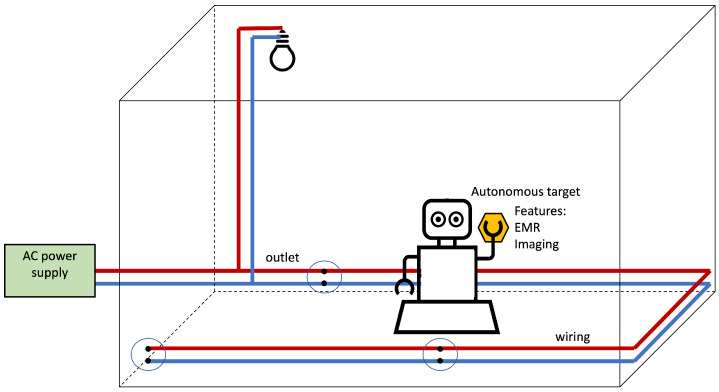
Infra-less Target-based PLPS architecture. The PLPS components are highlighted in yellow. In this scenario, the target uses the electrical network for positioning without PLPS infrastructure.

**Figure 3 sensors-22-07827-f003:**
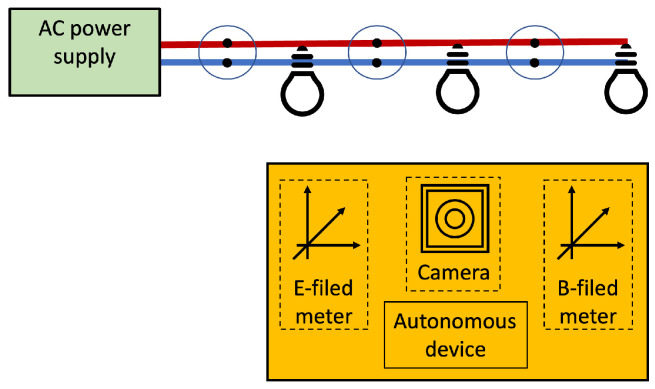
Sensors employed for self-positioning by some Infra-less Target-based PLPS.

**Figure 4 sensors-22-07827-f004:**
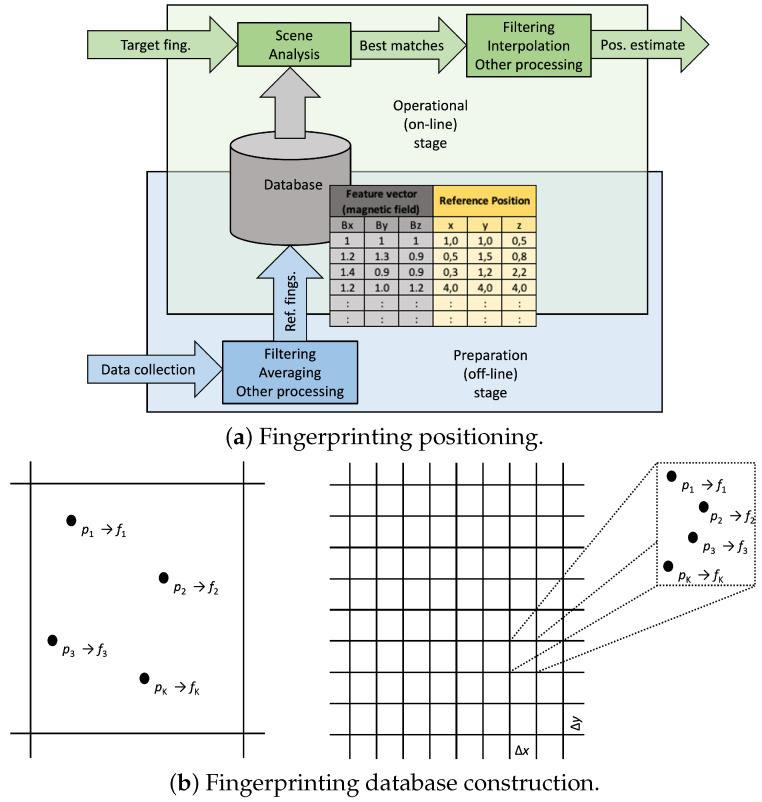
Fingerprinting positioning consists of two stages. During preparation, the database is collected. The feature vector comprises the AC EMR in each axis. In the operational phase, the scene analysis algorithm returns position candidates for the target position. At the bottom, one sees that a fingerprint may contain the positioning features collected at one position (left) or combine the features collected within a cell.

**Figure 5 sensors-22-07827-f005:**
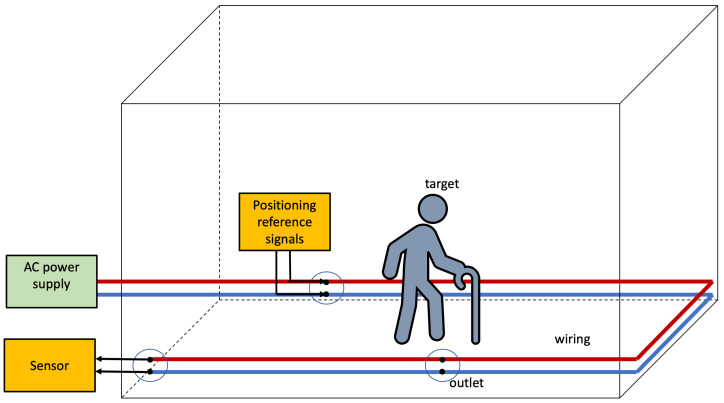
Infra-based and Target-obtuse unobtrusive PLPS architecture. The PLPS components are highlighted in yellow. In this scenario, the system estimates the position of the target whilst the target is unaware.

**Figure 6 sensors-22-07827-f006:**
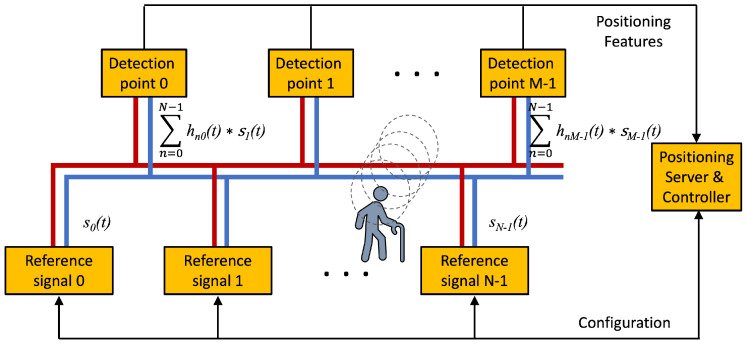
Basic scheme for Infra-based Target-obtuse PLPS.

**Figure 7 sensors-22-07827-f007:**
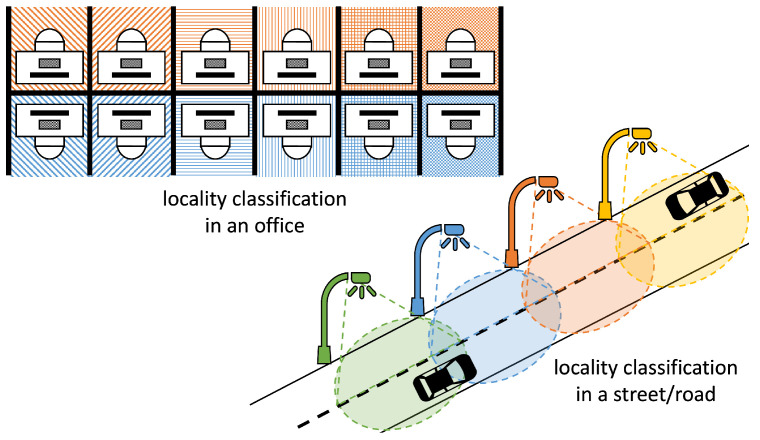
Examples of locality classification systems.

**Figure 8 sensors-22-07827-f008:**
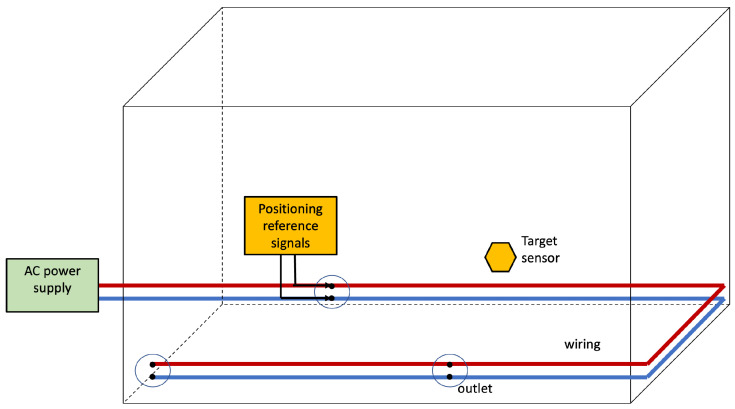
Infra-based and Target-based PLPS architecture. The PLPS components are highlighted in yellow. In this scenario, the electrical network and the target collaborate to obtain the position estimate.

**Figure 9 sensors-22-07827-f009:**
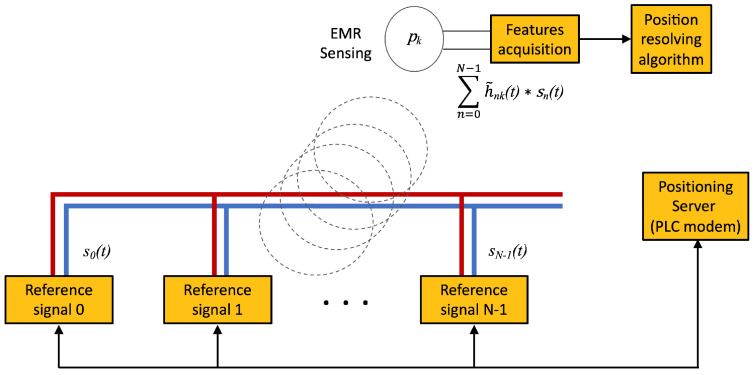
Basic scheme for Infra-based Target-based PLPS.

**Figure 10 sensors-22-07827-f010:**
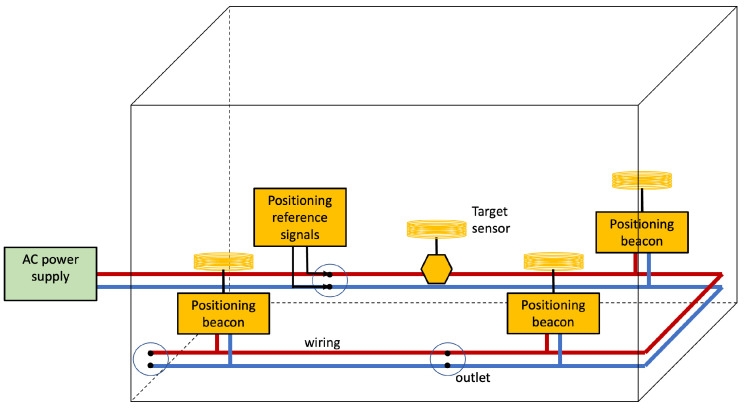
IBTB PLPS architecture using magnetic beacons. The PLPS components are highlighted in yellow. The electrical network delivers the positioning beacon to the anchor coils.

**Figure 11 sensors-22-07827-f011:**
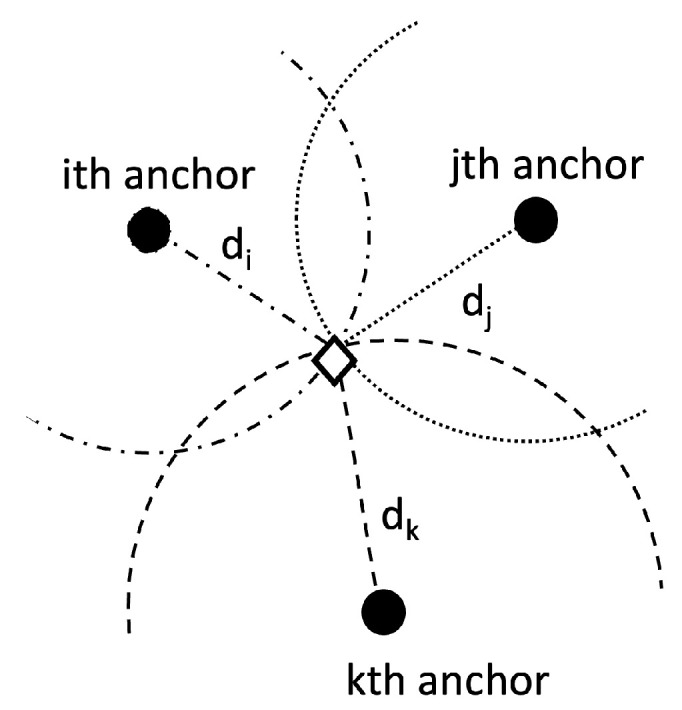
Geometrical model for range-based multilateration. To resolve the position of the target (represented by the rhomboid), one must find where the circles intercept.

**Figure 12 sensors-22-07827-f012:**
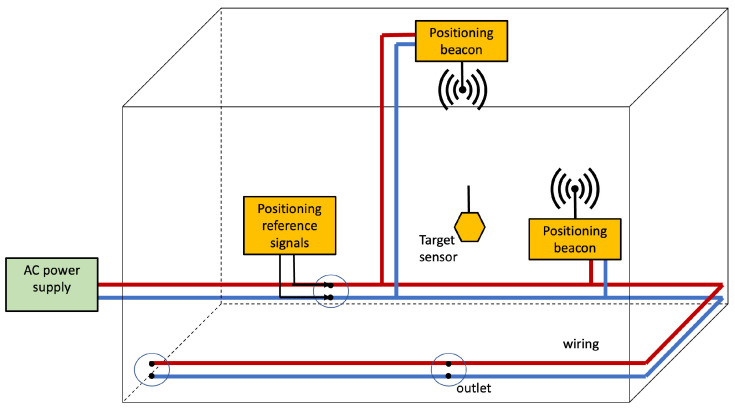
Infra-based and Target-based PLC-Wireless PLPS architecture. The PLPS components are highlighted in yellow. The electrical network delivers the positioning beacon to the anchors. Although the figure shows two positioning anchors that emit the RF beacons for positioning, more may be employed.

**Figure 13 sensors-22-07827-f013:**
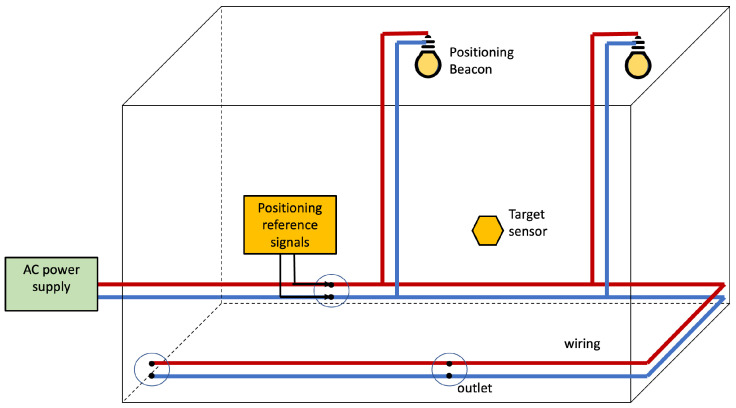
Infra-based and Target-based PLC–VLC PLPS architecture. The PLPS components are highlighted in yellow. The electrical network delivers the positioning beacon to the light anchors for transmission.

**Table 1 sensors-22-07827-t001:** Categories, characteristics, and broad application scenarios of positioning systems using the mains. The category indicates if the PLPS employs specific infrastructure to produce and inject reference signals or not (based × less) and if the target is involved in the positioning process or not (based × obtuse). The characteristics specify if and where the reference signals are produced, if they may come from PLC, which system part collects the positioning features, and where is the position resolved. The application scenarios are self-, unobtrusive, and transparent positioning.

Category	Characteristics					Application Scenarios		
	Reference Signals		Sensor Location	Position Computation		Self-Positioning	Unobtrusive	Transparent
	Exist	PLC-born		Where	Cooperation			
**Infra-less Target-based** (ILTB)	No	No	Target	Target	No	Yes	Not Applicable	Not Applicable
**Infra-based Target-obtuse** (IBTO)	Yes	Possible	Infra	Infra	No	No	Yes	Possible
**Infra-based Target-based** (IBTB)	Yes	Possible	Target	Infra or Target	Possible	No	No	Yes

**Table 2 sensors-22-07827-t002:** List of PLPS and experiments with results indicating possibilities for PLC-IPS design.

Purpose	Category	Feature Principle	Feature Source	Instrument/Sensor	Reach	Utilization	PLC Band	Ref.
Robot plugging-in (Self-positioning)	ILTB	EMR	AC carrier, LV	E-filed detector	cm	Outlet detection from the E-field	None	[42]
Robot plugging-in (Self-positioning)	ILTB	Image processing	Outlet Image	Camera	mm	Outlet alignment using image processing	Not applicable	[42]
Robot plugging-in (Self-positioning)	ILTB	EMR	AC carrier, LV	Copper foil	cm	Outlet detection detection from the E-field	None	[32]
Robot plugging-in (Self-positioning)	ILTB	EMR	AC carrier, LV	Ground prong	mm	Outlet alignment detection using the E-field	None	[32]
UAV perching (Self-positioning)	ILTB	EMR	AC carrier, MV	Magnetometer	cm and m	Relative distance	None	[43,44]
SLaM (Self-positioning)	ILTB	EMR	AC carrier, LV	Tank circuit and microphone	m	Fingerprint positioning	None	[45]
Locality classification (Unobtrusive)	IBTO	Body-wiring coupling	70 MHz carrier	Voltage peak detector	m	Fingerprint positioning	Greater than BB-PLC	[35,46]
IPS	IBTB	EMR	30 and 447 kHz carriers	Tank circuit	m	Fingerprint positioning	NB-PLC	[24]
IPS	IBTB	EMR	44 carriers (0.447–20 MHz)	Loop antenna of 1 m	m	Fingerprint positioning	MB-PLC BB-PLC	[47]
IPS	IBTB	Induced voltage	24, 125, 189 kHz	Coils and resonators	cm and m	Ranging and multilateration	NB-PLC	[48,49,50]
Eavesdropping	None (IBTB)	EMR	1–40 MHz band	Loop antennas of 10 and 30 cm	m	Channel response (CSI)	MB-PLC BB-PLC	[51,52]
Capacity evaluation	None (IBTB)	EMR	1.7–100 MHz band	Monopole antenna	m	Channel response (CSI)	MB/BB-PLC	[53]
Capacity evaluation	None (IBTB)	EMR	10–30 MHz band	Coupled inductors	cm	Channel response (CSI)	BB-PLC	[54]

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
