# Peer review of "On Power Line Positioning Systems"

_sensors, 2022, doi:10.3390/s22207827_

Round 1

Reviewer 1 Report

The reviewed article presents in an extensive way methods which can be used in power line positioning field. The paper has an overview nature, taking into account some aspects of PLC (Power Line Communication) systems. Although the presented methods are interesting, there is little bit a lack of detailed data that would allow to significantly explore the each topic. The following questions/remarks/suggestions arose to me during the review of the paper:

1. Could the authors show clearly in the introduction section the contribution of the paper?

2. The article is long, so the authors did not avoid minor typographical errors.

  - in line 278, a dot appeared before the end of the sentence,

  - in line 362 is "KHz" and it should be "kHz",

- to better understand the enumeration in line 528, I suggest changing from "to [120] to i) fulfill" to "to [120] to: i) fulfill" (the colon is missing in the current version of the work).

3. In the Infra-based Target-based PLPS method, tones within the range that are audible to the human ear are used. Does the human hear them? Some sentence concerning this phenomenon should be added to 5.1. section.

4. Some of the methods discussed in the work use electromagnetic fields of different frequencies that affect the human body. It would be good to add some remarks regarding the influence of these fields on the human organism.

Author Response

Answer to Reviewer 1

The reviewed article presents in an extensive way methods which can be used in power line positioning field. The paper has an overview nature, taking into account some aspects of PLC (Power Line Communication) systems. Although the presented methods are interesting, there is little bit a lack of detailed data that would allow to significantly explore the each topic.

Answer: Thank you very much for your comments and careful review.

The following questions/remarks/suggestions arose to me during the review of the paper:

  1. Could the authors show clearly in the introduction section the contribution of the paper?

Action: We introduced new text in the Introduction, trying to better explain the contribution of this review paper. Please see lines 56 to 64, 78, and 79 in the revised manuscript.

  1. The article is long, so the authors did not avoid minor typographical errors.

  - in line 278, a dot appeared before the end of the sentence,

Action: Thank you for finding this typo. We corrected it and searched for similar typos throughout the manuscript, see line 294 in the revised manuscript.

  - in line 362 is "KHz" and it should be "kHz",

Action: Thank you for finding this typo. We corrected it and searched for similar typos throughout the manuscript, see line 383 in the revised manuscript.

- to better understand the enumeration in line 528, I suggest changing from "to [120] to i) fulfill" to "to [120] to: i) fulfill" (the colon is missing in the current version of the work).

Action: We have altered the text as suggested. Please see line 616 in the revised manuscript.

  1. In the Infra-based Target-based PLPS method, tones within the range that are audible to the human ear are used. Does the human hear them? Some sentence concerning this phenomenon should be added to 5.1. section.

Action: We revised Section 5.1 to clarify the infra-based target-based PLPS strategy and the problem of collecting the features in this arrangement. Please see lines 374-380 in the revised manuscript.

  1. Some of the methods discussed in the work use electromagnetic fields of different frequencies that affect the human body. It would be good to add some remarks regarding the influence of these fields on the human organism.

Action: We have added new text in the second paragraph of Section 6 in the revised manuscript addressing this issue. Please see lines 544-549 in the revised manuscript.

Reviewer 2 Report

The paper investigates and summarizes the power line positioning system, and divides the power line positioning system into three categories: ILTB, IBTO and IBTB according to whether the infrastructure is required to generate reference positioning signals and whether the positioning target needs to collect positioning signals, and introduces their working principles and application directions respectively. This paper focuses on the positioning system of power line infrastructure, which has good application value. However, there are still some problems as follows:

A. Section 3-5 of the paper only introduce the working principle and application direction of ILTB, IBTO and IBTB power line positioning systems, without comparing the advantages and disadvantages of existing methods horizontally and lacking the author's comments and thoughts. 

B. It is suggested to supplement the discussion on the role of power line infrastructure in power line positioning technology, such as the characteristics of positioning signals provided, how to collect positioning signal information from power line infrastructure, and how to make power line infrastructure generate reference positioning signals. 

C. Section 5 of the original mentioned "inject positioning purpose tones in the building's electrical wiring at different locations". Can you explain Positioning purpose tones?

D. There are some errors in the contents of the chart in the article, for example: 1) PLPS system is classified into three categories in Table 1 -- "ILTB", "IBTO", and "IBTB", but in the line of position classification and purpose in Table 2, it is classified as "ILTO", should it be "IBTO"? 2) Table 1 header is not centered, which affects reading. 3) In Figure 2, EMR and Imaging do not have sensors but features.

E. There are some problems in the text format and spelling. For example, the full name of A Power Line Positioning System (PLPS) is not used for the first time.

Author Response

Answer to Reviewer 2

The paper investigates and summarizes the power line positioning system, and divides the power line positioning system into three categories: ILTB, IBTO and IBTB according to whether the infrastructure is required to generate reference positioning signals and whether the positioning target needs to collect positioning signals, and introduces their working principles and application directions respectively. This paper focuses on the positioning system of power line infrastructure, which has good application value. However, there are still some problems as follows:

Answer: Thank you very much for your comments and careful review.

Section 3-5 of the paper only introduce the working principle and application direction of ILTB, IBTO and IBTB power line positioning systems, without comparing the advantages and disadvantages of existing methods horizontally and lacking the author's comments and thoughts.

Action: We added new text in Section 6 of the revised manuscript for further cross-comparison. Please see lines 540-576 of the revised manuscript.

It is suggested to supplement the discussion on the role of power line infrastructure in power line positioning technology, such as the characteristics of positioning signals provided, how to collect positioning signal information from power line infrastructure, and how to make power line infrastructure generate reference positioning signals.

Action: While Section 2 discusses the three PLPS architectures (characteristics and properties), Sections 3, 4, and 5 discuss the specificities of the different architectures. In the revised manuscript, to improve the points indicated by the reviewer, we create subsections presenting the system model, the positioning references, their collection, and their use for positioning in the sections discussing each of the three architectures. Please see the new subsection titles in sections 3, 4, and 5. Besides, we also introduced text in the manuscript in lines 56-64 in the Introduction and in lines 171-177 to highlight these aspects for the readers.

Section 5 of the original mentioned "inject positioning purpose tones in the building's electrical wiring at different locations". Can you explain Positioning purpose tones?

Action: We tried to improve the presentation by explaining the concept in the first paragraph of Section 5.1. Please see lines 374-380 in the revised manuscript. This is also better explained for infra-based target-obtuse PLPS in the new subsections 4.2 and 4.3 and for and infra-based target-based PLPS in the new subsections 5.2 and 5.3.

There are some errors in the contents of the chart in the article, for example: 1) PLPS system is classified into three categories in Table 1 -- "ILTB", "IBTO", and "IBTB", but in the line of position classification and purpose in Table 2, it is classified as "ILTO", should it be "IBTO"? 2) Table 1 header is not centered, which affects reading. 3) In Figure 2, EMR and Imaging do not have sensors but features.

Action:

  • Thank you for finding the ILTO typo. We corrected it and searched for similar typos throughout the manuscript. Please see the changes in Table 1 in the revised manuscript.
  • Regarding the table header, we use the latex format provided by MDPI, and we do not control where the table caption is positioned.
  • Please see the corrected Figure 2 on page 7 in the revised manuscript.

There are some problems in the text format and spelling. For example, the full name of A Power Line Positioning System (PLPS) is not used for the first time.

Action: We tried to better introduce the term; see the modified text in line 4 in the Abstract and in lines 44-47 in the Introduction Section.

Reviewer 3 Report

This article tries to  present how the power-line infrastructure can be used for positioning. Some basic PLC positioning methods and their applications are introduced. But the new technology and method of positioning with PLC were not explained clearly. Some references cited are not closely related to PLC positioning applications, and the literature is relatively old. 

Author Response

Answer to Reviewer 3

This article tries to  present how the power-line infrastructure can be used for positioning. Some basic PLC positioning methods and their applications are introduced. But the new technology and method of positioning with PLC were not explained clearly. Some references cited are not closely related to PLC positioning applications, and the literature is relatively old. 

Answer: Thank you very much for your comments and careful review. This is a review paper to organize the initiatives for power line-based positioning. These initiatives have had a non-constant development over time; the power lines positioning systems references reflect that. Besides, we present references to link PLPS to applications and to other positioning techniques and methods; other references are intended to present how PLC communication technology, its requirements, and the effects it produces that can make it useful for positioning.

Action: We tried to improve the manuscript's quality and detail the possible techniques to deploy positioning systems using power lines. Please, see the new organization for the sections on infra-based target-obtuse and infra-based target-based PLPS, in Section 4 and Section 5, respectively. Aiming at a better comparison among the different system parts in the different architectures, divided sections 3, 4, and 5 into subsections presenting the system model, positioning features, their use for resolving the position, and their collection. Please see the new structures of sections 3, 4, and 5.

Reviewer 4 Report

Congratulations on the publication, it is interesting and contains an extensive review of the literature and a precise analysis of real situations. For the sake of content, please consider positioning your publication in the review type. To my full happiness, I miss the list of abbreviations used.

Author Response

Answer to Reviewer 4

Congratulations on the publication, it is interesting and contains an extensive review of the literature and a precise analysis of real situations. For the sake of content, please consider positioning your publication in the review type. To my full happiness, I miss the list of abbreviations used.

Action: Thank you very much for your comments and careful review. For submission, we have positioned our paper as a review. Regarding the abbreviation list, following the MDPI format, it is placed after the Conclusion section.

Round 2

Reviewer 2 Report

The author answers and modifies the questions raised in the modification opinions. In Section 3-5, the sub-chapter "Positioning Features" was added, and the signal characteristics of ILTB, IBTO and IBTB power line Positioning systems applications as well as the relationship between signal characteristics and power line facilities were introduced, which made the content more detailed and credible. According to the modification opinion, the author adds an explanation of the occurrence of 'tones' in Section V, that is,' sinusoidal waves', which eliminates the ambiguity of the article and facilitates the reader's understanding; In Section 6, the author adds a comparison of the advantages and disadvantages of ILTB, IBTO, and IBTB power line positioning systems from the application level. In addition to the horizontal comparison at the application level, it is suggested that the author add a comparison and analysis of the accuracy and positioning performance of different types of power line positioning systems. In addition, the ILTB and IBTO system models introduced in this paper are similar to the positioning system based on leaked coaxial cable. It is suggested that the author compare the above models and add the advantages of the system introduced in this paper.

Author Response

Comment:

The author answers and modifies the questions raised in the modification opinions. In Section 3-5, the sub-chapter "Positioning Features" was added, and the signal characteristics of ILTB, IBTO and IBTB power line Positioning systems applications as well as the relationship between signal characteristics and power line facilities were introduced, which made the content more detailed and credible. According to the modification opinion, the author adds an explanation of the occurrence of 'tones' in Section V, that is,' sinusoidal waves', which eliminates the ambiguity of the article and facilitates the reader's understanding; In Section 6, the author adds a comparison of the advantages and disadvantages of ILTB, IBTO, and IBTB power line positioning systems from the application level.

Answer: We thank the reviewer for the careful review.

Comment:

In addition to the horizontal comparison at the application level, it is suggested that the author add a comparison and analysis of the accuracy and positioning performance of different types of power line positioning systems.

Action: We have introduced text in 582-596 to discuss this topic.

Comment:

In addition, the ILTB and IBTO system models introduced in this paper are similar to the positioning system based on leaked coaxial cable. It is suggested that the author compare the above models and add the advantages of the system introduced in this paper.

Action: We have introduced text in lines 419-424 to mention this.

Reviewer 3 Report

Thanks for the authors's response. The manuscript has been sufficiently improved,and all the questions have been explained clearly. 

Author Response

Comment:

Thanks for the authors's response. The manuscript has been sufficiently improved,and all the questions have been explained clearly. 

Answer: We thank the reviewer for the careful review.